# Microtubule Association of EML4–ALK V3 Is Key for the Elongated Cell Morphology and Enhanced Migration Observed in V3 Cells [note 1]

**DOI:** 10.3390/cells13231954

**Published:** 2024-11-25

**Authors:** Savvas Papageorgiou, Sarah L. Pashley, Laura O’Regan, Kees R. Straatman, Andrew M. Fry

**Affiliations:** 1Department of Molecular and Cell Biology, University of Leicester, Lancaster Road, Leicester LE1 7RH, UK; sarah.pashley@nottingham.ac.uk (S.L.P.); laura.oregan@bristol.ac.uk (L.O.); andrew.fry@leicester.ac.uk (A.M.F.); 2Advanced Imaging Facility, Core Biotechnology Services, University of Leicester, Leicester LE1 9HN, UK; krs5@leicester.ac.uk

**Keywords:** EML4–ALK, therapy, microtubules, morphology, migration, NEK9, NEK7

## Abstract

The EML4–ALK oncogene drives tumour progression in approximately 5% of cases of non-small-cell lung cancers. At least 15 EML4–ALK variants have been identified, which elicit differential responses to conventional ALK inhibitors. Unfortunately, most, if not all, patients eventually acquire resistance to these inhibitors and succumb to the disease, which warrants the need for alternative targets to be identified. The most aggressive variant, EML4–ALK variant 3 (V3), assembles into a complex on interphase microtubules together with the NEK9 and NEK7 kinases, which leads to the downstream phosphorylation of NEK7 substrates. Overall, this promotes an elongated cell morphology and an enhanced migratory phenotype, which likely contributes to the increased metastasis often seen in V3 patients. Here, using two separate approaches to displace V3 from microtubules and a variety of in vitro assays, we show that microtubule association of EML4–ALK V3 is required for both V3 phenotypes, as removal of the oncogenic fusion protein from microtubules led to the dissociation of the V3–NEK9–NEK7 complex and the reversal of both phenotypic changes. Overall, we propose that targeting the interaction between EML4–ALK V3 and microtubules might offer a novel therapeutic option, independent of ALK activity, for V3+ NSCLC patients with acquired resistance to ALK inhibitors.

## 1. Introduction

Lung cancer remains the most commonly diagnosed cancer worldwide (12.4%) with currently the highest mortality rates (18.7%) [1]. It is histologically classified into small-cell lung cancer (SCLC) (15%) or non-small-cell lung cancer (NSCLC) (85%). The latter can be further subcategorized into adenocarcinomas, squamous-cell carcinomas and large-cell carcinomas [2]. Several oncogenic drivers of NSCLC have been described, which allows for the development of targeted therapeutics and further molecular subtyping [3,4]. Currently, point mutations in *KRAS* and *EGFR* genes are the most common types of alterations in NSCLC followed by *BRAF* mutations and *ALK* gene rearrangements [5].

Echinoderm microtubule-associated protein-like 4 (EML4) is a microtubule-associated protein whose N-terminal domain (NTD) comprises a trimerization domain (TD) and an unstructured basic region. The EML4 C-terminal region is made up of a structured tandem atypical propeller (TAPE) domain which consists of a series of tryptophan–aspartate (WD) repeats and a hydrophobic motif in the EML protein (HELP) motif [6]. Echinoderm microtubule-associated protein-like 4 (EML4)–anaplastic lymphoma kinase (ALK) is an oncogenic fusion protein that results from an inversion on the short arm of chromosome 2, leading to the expression of a constitutively active tyrosine kinase which drives tumour progression in 5% of cases of NSCLC [7]. Differential *EML4* breakpoints give rise to distinct EML4–ALK variants which all express the ALK kinase domain but different EML4 regions [8]. Variants expressing incomplete EML4 TAPE domains are known as “long variants”, whereas those that lack this domain entirely are called “short variants”. Currently, more than 15 variants have been identified; however, variants 1 to 5 (V1–V5) are the most prevalent in patients [8]. Specifically, the long variant, V1 (33%), and the short variant, V3 (29%), are the most common in patients. EML4–ALK V3 is also correlated with the most aggressive disease and a high level of metastasis in patients [7,9,10]. 

Specific short variants, such as EML4–ALK V3, are localized to the microtubule network, whereas long variants, such as V1, appear to have a diffuse cytoplasmic localization [11,12]. This is due to the domain organization of these variants, which appears to be important for the localization of the protein. Specifically, variants expressing the TD and basic region of EML4 can bind to microtubules, whereas variants lacking either one of these two regions cannot associate with microtubules [11]. Interestingly, of the five most common EML4–ALK variants, only V3 is localized to microtubules. V5 lacks the basic region of EML4, preventing its association with microtubules. Although long EML4–ALK variants possess both regions necessary for microtubule binding, they also express incomplete parts of the EML4 TAPE domain, which specifically perturbs the interaction of these variants with microtubules [11,12]. 

ALK inhibitors represent the standard treatment for ALK+ patients with a plethora of first-, second- and third-generation inhibitors already being used in the clinic. However, despite initial responses, patients eventually relapse and develop resistance [7]. Interestingly, the response to ALK inhibitors appears to be partly dependent on which EML4–ALK variant is present. For example, V3 confers a higher resistance to ALK inhibitors compared to V1, as cells expressing V1 were more sensitive to 6 different ALK inhibitors compared to cells expressing V3 [13]. Overall, this warrants the need for alternative targets to be identified to bypass the need for these inhibitors, particularly for EML4–ALK V3+ NSCLC patients. 

A novel pathway has recently been identified which involves the recruitment of an EML4–ALK V3–NEK9–NEK7 complex to interphase microtubules. Cells expressing this complex appear to have stabilized microtubules, evident by increased levels of acetylated tubulin, while also having an elongated morphology and enhanced migration [14]. Both phenotypic changes are independent of ALK catalytic activity, as the addition of ALK inhibitors did not lead to the reversal of either phenotype. However, they are dependent on NIMA-related kinases 9 (NEK9) and 7 (NEK7), as depletion of either kinase led to the reversal of both phenotypes [14]. Interestingly, Pashley et al. (2024) have demonstrated that at least the elongated cell morphology is also dependent on NEK7 phosphorylation of the motor protein, Eg5-S1033 [15]. Therefore, we hypothesized that assembly of the V3–NEK9–NEK7 complex to microtubules likely leads to the untimely activation of the NEK7 kinase. This in turn promotes the premature recruitment of NEK7 substrates to interphase microtubules, resulting in their subsequent phosphorylation by NEK7 and promotion of the altered phenotypes.

Here, we examined whether the microtubule interaction of EML4–ALK V3 is a pre-requisite for driving the V3 phenotypes. If so, targeting this interaction may prove to be a novel therapeutic approach alternative to ALK catalytic activity for V3+ patients. Overall, using the “knocksideways” approach to relocate V3 from microtubules to the mitochondrial membrane, we show that displacing EML4–ALK V3 from microtubules suppresses the elongated cell morphology and the migratory and invasive potential. Importantly, consistent results were also obtained using phosphomimetic mutants of EML4–ALK V3 that associate less with microtubules compared to wild-type V3. Finally, we show that in the absence of EML4–ALK V3 from microtubules, the V3–NEK9–NEK7 complex dissociates, resulting in the reversal of both phenotypes. Overall, our data suggest that microtubule recruitment of EML4–ALK V3 is required for the generation of the elongated morphology and enhanced migration observed in V3 cells. V3 displacement from microtubules likely prevents the activation of NEK7 and the subsequent phosphorylation of its downstream targets, leading to the reversal of both phenotypic changes. This raises the future prospect of targeting the association of EML4–ALK V3 with microtubules as an alternative therapeutic route for V3+ NSCLC patients.

## 2. Materials and Methods

### 2.1. Plasmid Construction and Site-Directed Mutagenesis

To generate the triple phosphomimetic mutant of EML4–ALK V3 (S134D/S144D/S146D), a double phosphomimetic V3 mutant (S144D/S146D) subcloned into a pcDNA3.1 Hygro (Invitrogen, Waltham, MA, USA) vector was used as a template. Q5 site-directed mutagenesis (New England Biolabs, Ipswich, MA, USA) was performed, followed by XL-10 GOLD *E. coli* bacterial transformation. The following primer sequences were used: 5′-ggaatctcatgataatgatcaaagtcc-3′ and 5′-cctgtctctcttttttttct-3′. Both mutant constructs were confirmed by next generation sequencing (Source Bioscience, Nottingham, UK).

To generate the “knocksideways” FKBP–V3 construct, a bicistronic plasmid (Addgene, plasmid #128267) was used as a template, which encodes an mCherry–FRB domain with a MitoTrap and an FKBP–GFP nanobody [16]. The following primer sequences were used to remove the GFP nanobody: 5′-tgaggaggcgcggcggcagc-3′ and 5′-cctttctagacctccgcgtgtc-3′. Full-length wild-type EML4–ALK V3 cDNA was isolated by PCR from a plasmid previously generated in our lab [14] using the following primer sequences: 5′-tctggaggcgcacagatggacggtttcgccggc-3′ and 5′-cgtacttggtcggacccgggactactcctccgcgccgc-3′. Gel-purified PCR products underwent ligation using the InFusion kit (Takara, cat# 638947, Kusatsu, Shiga), followed by amplification in DH5α bacteria. The size of the construct was confirmed by agarose gel electrophoresis, and the expression of the appropriate proteins was confirmed by immunofluorescence (IF) microscopy and Western blot (WB) analyses.

### 2.2. Cell Culture, Drug Treatments and Transfection

Beas2B bronchial endothelial and U2OS osteosarcoma parental cells were grown in an RPMI (Gibco, Grand Island, NY, USA) or DMEM (Gibco, Grand Island, NY, USA) medium respectively, supplemented with 10% (*v*/*v*) Foetal bovine serum (FBS) and 1X Penicillin–Streptomycin (PS). Doxycycline-inducible U2OS cells expressing activated myc-NEK9 (ΔRCC1) (aNEK9) were grown in a DMEM medium supplemented with 10% (*v*/*v*) FBS, 1X PS and 800 ng/mL of puromycin. Doxycycline-inducible HeLa cervical carcinoma cells expressing activated YFP–NEK7 (Y97A) (aNEK7) or YFP–NEK7 kinase dead (D179N) (NEK7 KD) were grown in a DMEM medium supplemented with 10% (*v*/*v*) FBS, 1X PS and 200 μg/mL of hygromycin. The expression of aNEK9, aNEK7 or NEK7 KD was induced with 1 μg/mL of doxycycline for 72 h. All cells were grown in a 37 °C/5% CO_2_ environment and kept in a culture for a maximum of two months. They were stored in liquid nitrogen at a low passage number and underwent monthly mycoplasma testing using an in-house PCR-based protocol. For the phosphomimetic constructs, 2 μg of plasmid DNA was used together with 3 μL of the FuGENE HD transfection reagent (Promega, Madison, WI, USA) to transiently transfect cells, according to the manufacturer’s instructions. For the “knocksideways” construct, 1 μg of plasmid DNA was used together with 4 μL of FuGENE instead. To maintain the expression of the “knocksideways” construct in stable U2OS cell lines, the growth medium was supplemented with 1.6 mg/mL of the G418 antibiotic. Monoclonal cell lines were generated by selecting clones that exhibited a sufficient expression of the desired proteins on a Western blot and by an immunofluorescence analysis. Rapamycin was prepared at 250 mg/mL in DMSO and used at a final concentration of 200 nM. The period of the treatment with rapamycin is indicated in the figure legends.

### 2.3. Preparation of Cell Extracts, SDS-PAGE and Western Blotting

All cells were lysed in a RIPA lysis buffer (50 mM of Tris-HCl, pH 8.0, 150 mM of NaCl, 1% (*v*/*v*) SDS, 0.5% (*w*/*v*) sodium deoxycholate, 0.5% (*v*/*v*) Nonidet P-40, and 0.5% (*v*/*v*) Triton X-100) supplemented with a 1X protease cocktail inhibitor (PIC), 1 mM of DTT, 30 μg/mL of DNAse and 30 μg/mL of RNAse A, prior to analysis by SDS-PAGE and Western blotting. The primary antibodies used were against ALK (1:1000, cat# 3633, Cell Signalling Technology, Danvers, MA, USA); GFP (1:1000, ab6556, Abcam, Cambridge, UK); FKBP12 (1:500, PA1-026A, Invitrogen); mCherry (1:1000, A85306, Antibodies.com, Cambridge, UK); NTD-EML4 (1:500, A301-908A, Invitrogen); NEK9 (1:200, sc-100401, Santa Cruz Biotechnology, Dallas, TX, USA); myc (1:500, 9B11, Cell Signalling Technology) and GAPDH (1:1000, 2118, Cell Signalling Technology). The secondary antibodies used were anti-rabbit (1:2000, A150-102, Bethyl Laboratories, Montgomery, TX, USA) or anti-mouse (1:2000, A90-116P, Bethyl Laboratories) horse radish peroxidase (HRP)-labelled IgGs. Western blots were detected using enhanced chemiluminescence (ECL; Pierce, Appleton, WI, USA) and developed using the GeneGnome chemiluminescence imaging system (Syngene, Bangalore, India).

### 2.4. Immunoprecipitation

Cells were harvested and lysed in an IP buffer (0.01 M of Tris-HCl, pH 7.4, 0.05 M of NaCl, 5 mM of EDTA, 0.1 mM of sodium orthovanadate, 30 mM of sodium pyrophosphate, 50 mM of NaF, 1% (*v*/*v*) NP-40, and 0.1% (*w*/*v*) BSA) supplemented with a 1X protease cocktail inhibitor (PIC), 1 mM of DTT, 30 μg/mL of DNAse and 30 μg/mL of RNAse A. Lysates were pre-cleared with protein A/G magnetic beads (Pierce) for 1 h at 4 °C, before the proteins were immunoprecipitated overnight at 4 °C with antibodies against NEK9 (2 μg, sc-100401, Santa Cruz Biotechnology); ALK (1:100, cat# 3633, Cell Signalling Technology) or IgG (2 μg, cat# 12-370, Sigma, St. Louis and Burlington, MA, USA).

### 2.5. Fixed- and Live-Cell Microscopy

For fixed-cell microscopy, cells grown on acid-etched coverslips were fixed and membrane-permeabilized in ice-cold methanol for 20 min at −20 °C. Cells were blocked in 3% (*w*/*v*) Bovine serum albumin (BSA) and 0.2% (*v*/*v*) Triton X-100 before incubation with primary antibodies diluted in PBS containing 3% (*w*/*v*) BSA and 0.2% (*v*/*v*) Triton X-100. The primary antibodies used were anti-GFP (1:500, ab6556, Abcam); anti-α-tubulin (1:1000, T5168, Sigma); anti-α-tubulin (1:1000, Invitrogen); anti-MTC02 (1:200, MAB1273, Sigma); anti-mCherry (1:500, A85306, Antibodies.com); anti-ALK (1:200, cat# 3633, Cell Signalling Technology); anti-NEK7 (1:500, SAB2501398, Sigma); anti-NEK9 (1:100, sc-100401, Santa Cruz Biotechnology); anti-NEK9 (1:100, HPA001405, Atlas Antibodies, Stockholm, Sweden); anti-Eg5 (1:200, 23333-1-AP, Proteintech Europe, Manchester, UK) and anti-FKBP12 (1:100, PA1-026A, Invitrogen). The secondary antibodies used were Alexa Fluor 488- and 594-conjugated donkey anti-rabbit or donkey anti-mouse (1:200, Invitrogen). Nuclei were stained with Hoechst 33258 (0.5 μg/mL, H3569, Invitrogen). 

For super-resolution microscopy, single optical sections were captured in SR mode on a Zeiss LSM980 Airyscan 2 microscope (Zeiss, Oberkochen, Germany) fitted with a Plan-Apochromat 63× oil objective (NA = 1.4) and further processed using the Zeiss Airyscan joint deconvolution module using 4 iterations. Lower-resolution confocal images were captured on a VisiTech Infinity 3 confocal microscope fitted with a Hamamatsu C11440-22CU Flash 4.0 V2 sCMOS camera, and a Plan Apo 20× objective (NA = 0.75) was used. For live-cell imaging, a PhaseFocus LiveCyte 2 label-free imaging system fitted with a PLN 10× objective (NA = 0.25) was used with phase imaging for wound healing and a laser power of 10% YFP (528–559 nm) or RFP (600–650 nm) LEDs for single-cell tracking experiments of cells expressing the phosphomimetic mutants or “knocksideways” construct, respectively.

### 2.6. Individual-Cell Tracking Assays

Cells were transiently transfected with the EML4–ALK V3 phosphomimetic mutants or the “knocksideways” construct, treated with rapamycin and subjected to live-cell imaging, as described above. Images were captured every 15 min for a total of 6 h. For each condition, the experiment was performed in triplicate, and tracks from 10 individual cells were measured per replicate (n = 30 cells per condition). The total distance covered (μm) and average velocity (μm/min) were calculated using the manual tracking plugin in Fiji [17].

### 2.7. Wound-Healing Migration Assays

U2OS parental cells or U2OS cells stably expressing the “knocksideways” construct were grown at 90–100% confluency in 6-well plates and treated with rapamycin, before a P200 pipette tip was used to scrape a 0.5–1 μm line across the centre of the well to generate the wound. Cells were washed three times in 1X PBS before adding fresh media and subjected to live-cell imaging for 12 h, with images captured hourly. Each experiment was performed in triplicate, and the cell-free area was measured in 10 distinct regions per condition (n = 30 regions per condition). The relative rate of wound healing and cell-free area at 6 and 12 h were calculated manually and compared using Fiji (v1.54f).

### 2.8. Three-Dimensional Spheroid Invasion Assays

Cells were seeded at 1 × 10^4^ cells /mL in an ultra-low attachment 96-well plate (Corning, cat# CLS4515, Corning, NY, USA) and allowed to grow for 72 h before embedding in 2.5 mg/mL of Matrigel (Corning, cat# CLS356234) for 1 h at 37 °C/5% CO_2_. Fresh media were then added to the cells before live-cell imaging for 72 h, with images being captured every 12 h. Cells treated with rapamycin were first seeded in a 100 mm plate and treated with 200 nM of rapamycin. These cells were then washed three times with 1X PBS to remove the rapamycin and transferred in an ultra-low attachment 96-well plate.

### 2.9. Quantification of Morphology

Beas2B cells transfected with either the phosphomimetic or “knocksideways” constructs were grown on acid-etched coverslips embedded in collagen. For this, acid-etched coverslips were covered in 1 mg/mL of rat tail (type I) collagen for 1 h, washed with 1X PBS 3 times and left to dry for 10 min at room temperature before cell seeding. Cells were then treated with rapamycin when needed before being fixed and processed for IF microscopy, as appropriate. Using Fiji, the total cell length (μm) and length of longest protrusions (μm) were manually measured in at least 30 cells in total per condition in three replicates.

### 2.10. Quantification of Co-Localization 

The JaCoP plugin [18] of Fiji was used to quantify co-localization. A detection threshold for each marker was manually set, and Manders’ co-localization (MC) coefficient was then calculated. Measurements were taken from 28–30 cells per condition, and each experiment was performed in triplicate.

### 2.11. Fluorescence Intensity Quantification

Low-resolution confocal images were analysed in Fiji. Measurements were taken from individual cells (n = 20 cells in total per condition) as well as the background surrounding those cells, and the fluorescence intensity was calculated using the following equation: integrated intensity area − (cell area * mean intensity of background). 

### 2.12. Statistical Analysis

All data were analysed using GraphPad PRISM (v9). Experiments with two sample groups were analysed with two-tailed unpaired *t*-tests, whereas those with more than two sample groups were analysed with one-way ANOVA with Tukey’s multiple comparisons tests. Individual measurements are shown together with the standard error of the mean (SEM), and each replicate is colour-coded. Statistical significance is indicated as follows: **, *p* = 0.0071 or 0.0020 or 0.0034; ****, *p* < 0.0001; ns, not significant.

## 3. Results

### 3.1. Generation and Characterisation of the “Knocksideways” FKBP:EML4–ALK V3

To study the importance of the microtubule localization of EML4–ALK V3, we generated a “knocksideways” construct encoding an FK506-binding protein (FKBP)-tagged V3. The “knocksideways” approach allows for the relocation of an FKBP-tagged protein from its native location within a cell to the mitochondria. This is due to the formation of a rapamycin-mediated complex between FKBP and a mitochondrially targeted FKBP–rapamycin binding (FRB) domain [19]. To confirm the localization of each protein, Beas2B parental cells were transiently transfected with the FKBP–V3 “knocksideways” or template bicistronic plasmids before fixation and staining with antibodies against FKBP, mCherry and a mitochondrial marker, MTC02 [20,21]. As expected, FKBP and FKBP–V3 in the template and new constructs, respectively, exhibited low co-localization levels (R = 0.19 and R = 0.21, respectively) with mCherry–FRB in the absence of rapamycin. Furthermore, mCherry–FRB strongly co-localized with MTC02 in both constructs (R = 0.83 and R = 0.81, respectively) (Figure 1A–C). Transfected cells were also collected, and lysates were prepared for Western blot analyses with antibodies against mCherry, FKBP, ALK and EML4 NTD. As expected, mCherry–FRB was detected at the same molecular weight (~40 kDa) in both constructs. Both the ALK and EML4 NTD blots revealed a strong band in cells transfected with FKBP–V3 at the expected size (~115 kDa). Interestingly, the EML4 NTD blot also revealed a faint band (~120 kDa) in non-transfected cells and cells expressing the template plasmid which likely represents the endogenous EML4. Finally, using an anti-FKBP antibody, we detected a single band in the template and FKBP–V3 samples (~25 kDa and ~115 kDa, respectively), as expected (Figure 1D,E). 

Next, to determine the time required for the mitochondrial relocation of FKBP–V3 to occur following treatment with rapamycin, a time-course experiment was performed in Beas2B cells transiently transfected with FKBP–V3. Cells were treated with 200 nM of rapamycin for 0–10 min before being fixed and stained for FKBP–V3 and mCherry–FRB. The staining pattern of FKBP–V3 appeared to be microtubule-like in untreated cells compared to the mitochondrial-like staining of mCherry–FRB (Figure 1F), suggesting that the addition of the FKBP tag to EML4–ALK V3 has no effect on the biological properties of the fusion protein. Within 10 s of treatment with rapamycin, the staining pattern of FKBP–V3 appeared to shift towards a mitochondrial-like pattern; however, microtubule-like staining was still evident. After 30 s of treatment with rapamycin, FKBP–V3 appeared to have a complete mitochondrial-like staining, with clear signs of co-localization (not measured) with mCherry–FRB (Figure 1F). Therefore, in all further experiments, Beas2B cells were treated with 200 nM of rapamycin for 5 min to ensure the complete relocation of FKBP:EML4–ALK V3 to mitochondria.

Overall, our data show that all proteins encoded within the “knocksideways” FKBP:EML4–ALK V3 construct are localized and behave as predicted while also being detected at their expected sizes on a Western blot. Finally, we show that the relocation of FKBP–V3 to the mitochondria in Beas2B cells occurs within 30 s following treatment with 200 nM of rapamycin.

### 3.2. Microtubule Recruitment of EML4–ALK V3 Is Required for the Generation of an Elongated Cell Morphology

Assembly of the EML4–ALK V3–NEK9–NEK7 complex on interphase microtubules leads to the generation of an elongated morphology and enhances cell migration [14]. To assess whether microtubule localization of EML4–ALK V3 is required for the development of an elongated morphology, the protrusion and total cell length of cells expressing FKBP–V3 were measured. Beas2B parental cells were transiently transfected with FKBP–V3 before being fixed and stained for ALK and α-tubulin. Transfected cells were also treated with 200 nM of rapamycin for 5 min and fixed 24 h post-treatment.

Prior to measuring the cell length, relocation of FKBP–V3 from microtubules to the mitochondria was confirmed by co-staining with antibodies against ALK and α-tubulin or mCherry. As expected, FKBP–V3 co-localized strongly to microtubules in untreated (R = 0.76) but not in treated (R = 0.12) cells. Upon the addition of rapamycin, higher co-localization between FKBP–V3 and mCherry–FRB was observed in treated (R = 0.76) compared to untreated (R = 0.20) cells (Figure 2A–C). Therefore, this confirmed the relocation of FKBP–V3 to the mitochondria due to the generation of the rapamycin-mediated complex between FKBP and FRB. An examination of the cell morphology indicated that parental cells lacking V3 had the shortest protrusions and total cell lengths (24.24 μm and 55.62 μm, respectively), whereas untreated cells expressing FKBP–V3 had longer protrusions and total cell lengths (57.13 μm and 97.69 μm, respectively) compared to parental cells. Following the relocation of FKBP–V3 to the mitochondria, rapamycin-treated cells exhibited both shorter protrusions and total cell lengths (33.67 μm and 63.14 μm, respectively) compared to untreated cells (Figure 2D–F).

Phosphomimetic mutants of EML4–ALK V3 (S144D/S146D or S134D/S144D/S146D) that associate less with microtubules compared to wild-type V3 were used as an alternative approach in assessing the importance of the microtubule association of V3 in driving both phenotypes. The residues chosen for mutagenesis were previously shown to be key in the microtubule association of the full-length EML4 protein [22,23]. Our co-localization analysis indicates that both the double (DM) and triple (TM) V3 mutants associate less with microtubules (R = 0.42 and R = 0.40, respectively) compared to the wild-type V3 (WT V3) (R = 0.64) (Appendix A). Western blot analyses also confirmed the expression of the desired proteins in transfected cells (Appendix A). The cell morphology results obtained using these mutants are consistent with those obtained with the “knocksideways” approach. Specifically, cells expressing WT V3 had the longest protrusions and total cell lengths (60.52 μm and 100.5 μm, respectively). Those expressing the DM or TM V3 had significantly shorter protrusions (39.79 μm and 34.86 μm, respectively) and total cell lengths (74.53 μm and 67.80 μm, respectively) compared to those expressing WT V3 (Appendix A–F).

Together, these data suggest that while cells adopt an elongated morphology in the presence of EML4–ALK V3 on interphase microtubules, displacement of V3 from the microtubule network prevents cells from acquiring this morphological phenotype.

### 3.3. Microtubule Recruitment of EML4–ALK V3 Is Required for Enhanced Cell Migration 

To assess whether microtubule recruitment of EML4–ALK V3 is necessary for enhancing cell migration, individual-cell tracking experiments were performed using the “knocksideways” approach and phosphomimetic mutants. The total distance (μm) covered over 6 h and the average velocity (μm/min) of the migrating cells were measured. Cells were also treated with 200 nM of rapamycin for 5 min and subjected to live-cell imaging 6 h post-treatment.

Beas2B parental cells covered the shortest distance and had the lowest average velocity (10.22 μm and 0.03 μm/min, respectively). Untreated cells expressing FKBP–V3 covered a significantly longer distance and had a higher average velocity (39.83 μm and 0.12 μm/min, respectively) compared to parental cells. Following the addition of rapamycin, cells covered a significantly shorter distance and had a lower average velocity (16.26 μm and 0.05 μm/min, respectively) compared to untreated cells (Figure 3A–C). Similar results were also obtained with the phosphomimetic V3 mutants, where cells expressing WT V3 covered the longest distance and had the highest average velocity (54.19 μm and 0.68 μm/min, respectively). Cells expressing either the DM or TM V3 covered a shorter distance (33.56 μm and 35.46 μm, respectively) and had a lower average velocity (0.34 μm/min and 0.33 μm/min, respectively) compared to cells expressing WT V3 (Appendix A–I).

To further assess the effects of the microtubule association of V3 on cell migration and invasion, U2OS cells stably expressing FKBP–V3 (i.e., U2OS:FKBP–V3 cells) were generated. Single clones (#3, #5, #6 and #9) were then further characterized by Western blot (Appendix A) and immunofluorescence analyses (Appendix A). Finally, the relocation of FKBP–V3 to the mitochondria following rapamycin treatment was also confirmed in two different clones. Interestingly, U2OS cells required at least 30 min of treatment with 200 nM of rapamycin for the relocation of FKBP–V3 to occur (Appendix A). Therefore, U2OS:FKBP–V3 cells were always treated with 200 nM of rapamycin for 2 h before being further processed. U2OS:FKBP–V3 cells were next used to examine their migratory and invasive potential, using a wound-healing and 3D spheroid invasion assays, respectively (Figure 3D–H). The results of the wound-healing assay suggest that U2OS parental cells had the lowest migration rate (slope = −0.05). Untreated U2OS:FKBP–V3 cells had the highest migration rate (slope = −0.08), which was significantly reduced following treatment with rapamycin (slope = −0.07) (Figure 3F). This was also consistent with an analysis of the relative cell-free area of each sample population at two different time points (Figure 3E). Specifically, the relative cell-free area at 6 and 12 h was the largest with parental cells (0.66 μm^2^ and 0.30 μm^2^, respectively). Untreated U2OS:FKBP–V3 cells had a smaller cell-free area at both time points (0.23 μm^2^ and 0.02 μm^2^, respectively) compared to parental cells, while rapamycin-treated cells had a larger cell-free area at both time points (0.48 μm^2^ and 0.12 μm^2^, respectively) compared to untreated U2OS:FKBP–V3 cells (Figure 3E). In the cell invasion assay, U2OS parental cells were unable to invade the Matrigel compared to cells expressing V3, demonstrating the oncogenic potential of EML4–ALK. U2OS:FKBP–V3 cells treated with rapamycin exhibited fewer and shorter invasive strands (74.46 μm) compared to untreated U2OS:FKBP–V3 cells (122.10 μm) (Figure 3G,H). 

Taken together, these data further confirm that the presence of EML4–ALK V3 on interphase microtubules enhances the cell migratory and invasive potential [14]. Importantly, displacing the V3 protein from the microtubule network decreased or completely inhibited the migratory and invasive properties of these cells.

### 3.4. Relocation of EML4–ALK V3 away from Microtubules Results in the Dissociation of the EML4–ALK V3–NEK9–NEK7 Complex

Both V3 phenotypes are dependent on NEK9 and NEK7 [14]; thus, we next examined the effect of V3 displacement from microtubules on the localization of both kinases. As expected, both NEK9 and NEK7 were recruited to microtubules (R~0.6 or 0.7, respectively) in the presence of FKBP–V3 compared to parental cells (R~0.4, for both). Most importantly, both NEK9 and NEK7 dissociated from microtubules (R~0.4 or 0.5, respectively) in the absence of V3 from microtubules in rapamycin-treated cells (Figure 4A–C). Next, we confirmed that both NEK9 and NEK7 co-localized strongly with EML4–ALK V3 (R~0.75 or 0.85, respectively) on microtubules in untreated cells. Following the addition of rapamycin, NEK9 dissociated from EML4–ALK V3 (R~0.2) and appeared to relocate to the cytoplasm. Surprisingly, although a reduced NEK7–V3 association was observed in rapamycin-treated cells, a majority of NEK7 remained strongly co-localized with V3 (R~0.75) on the mitochondria as well (Figure 4D–F), suggesting a possible NEK7–V3 interaction that is independent of NEK9. Together, this demonstrates that removal of EML4–ALK V3 from microtubules causes the release of both NEK9 and NEK7 from the microtubule network as well, explaining the reversal of both V3 phenotypes. By staining FKBP–V3 cells for NEK7 only, we observed a mitochondrial-like staining of NEK7 in rapamycin-treated compared to untreated cells, further supporting the relocation of NEK7 to the mitochondria following the displacement of V3 from microtubules to the mitochondrial membrane (Figure 4G). Finally, immunoprecipitation experiments confirmed the interaction of V3 with NEK9 on microtubules (i.e., untreated cells) and a loss of this interaction when V3 is removed from microtubules (i.e., rapamycin-treated cells) (Figure 4H).

As this is the first report of a possible NEK9-independent interaction between NEK7 and EML4–ALK V3, a series of control experiments were carried out to validate and confirm the results obtained above (Appendix A). Our data show that NEK7 staining in parental cells appears to be cytoplasmic irrespective of rapamycin treatment, suggesting that the drug treatment alone does not interfere with NEK7 localization (Appendix A). In addition, Beas2B cells expressing FKBP–V3 were co-stained for V3 and NEK7, and their staining patterns were closely examined to investigate any possible signs of bleed-through of the ALK fluorescence signal into the NEK7 channel. Although the two stains appeared similar, several regions within the cells were positive for ALK but not for NEK7. Furthermore, no signal was observed in the NEK7 channel (red) in cells solely stained for the V3 protein (Appendix A). Taken together, our data support that there is no bleed-through of the ALK fluorescence signal into the NEK7 channel and that NEK7 does indeed move to the mitochondria following the addition of rapamycin. 

Next, a series of control experiments were also performed to confirm the microtubule dissociation of NEK9 following the displacement of EML4–ALK V3 from microtubules. In agreement with the results obtained with the “knocksideways” approach, using the phosphomimetic V3 mutants, we demonstrate that NEK9 was recruited to microtubules (R~0.6) in the presence of WT but not the triple mutant (TM) YFP–V3 (R~0.43) compared to parental cells (R~0.45) (Appendix A). Moreover, NEK9 co-localized strongly with WT YFP–V3 in untreated and rapamycin-treated cells (R = 0.76 and R = 0.78, respectively), suggesting that rapamycin treatment alone does not interfere with NEK9 localization or its association with V3 (Appendix A). In addition, similar results were also obtained when staining with an alternative antibody (rabbit; cat#: HPA001405) that binds to a different region within the NEK9 sequence. Specifically, higher co-localization between NEK9 and FKBP–V3 was observed in untreated compared to rapamycin-treated cells (R~0.7 and R~0.4, respectively) (Appendix A). Finally, we set out to confirm the specificity of the mouse anti-NEK9 antibody that was originally used by means of a tetracycline-inducible U2OS cell line that expresses a constitutively active *myc*- NEK9 (ΔRCC1) (U2OS:aNEK9) [14]. U2OS:aNEK9 cells were induced with doxycycline for 0–72 h, and lysates were prepared for Western blot analyses using antibodies against *myc* and NEK9 (Appendix A). Both antibodies detected the ΔRCC1 NEK9 protein at ~75 kDa, while the NEK9 but not the *myc* antibody also detected the full length NEK9 protein at ~120 kDa (Appendix A). Cells induced with doxycycline for 72 h were then fixed and stained for NEK9 using the mouse anti-NEK9 antibody. As expected, the relative NEK9 intensity obtained was ~20-fold higher in doxycycline-induced cells compared to the one obtained in non-induced cells (Appendix A). Overall, these data confirm the specificity of the mouse antibody against the NEK9 protein and further validates the IF results shown in Figure 4A–E.

Together, these results suggest that in the absence of EML4–ALK V3 from the microtubule network, the V3–NEK9–NEK7 complex is disassembled, which leads to the reversal of both the increased cell length and enhanced migration, as shown in Figure 2 and Figure 3. Using the “knocksideways” approach, our data demonstrate that NEK7 relocates, together with V3, to the mitochondria, whereas NEK9 relocates to the cytoplasm. Interestingly, this suggests a possible NEK9-independent interaction between EML4–ALK V3 and NEK7.

### 3.5. Removal of EML4–ALK V3 from Microtubules Likely Prevents NEK7 Activation and Subsequent Phosphorylation of Eg5

NEK9 phosphorylates and activates both NEK6 and NEK7 by direct or allosteric-based mechanisms [24,25]. Therefore, assembly of the V3–NEK9–NEK7 complex on microtubules likely promotes NEK7 activation, which ultimately might lead to phenotypic changes. To determine whether NEK7 activation is required for the phenotypes described in this paper, HeLa cells expressing a doxycycline-inducible and constitutively active YFP–NEK7 (YFP–aNEK7) or NEK7 kinase dead (YFP–NEK7 KD) [14] were used. Cells were transfected with WT or TM V3 before being fixed and stained for α-tubulin or ALK to assess their morphology. Western blot analyses confirmed the expression of the mutant NEK7 proteins in both cell lines following a time-course (0–72 h) induction with doxycycline. Importantly, cell-length measurements, taken from within the same cells using both the α-tubulin and ALK stains, confirm that both stains can be used interchangeably to measure cell morphology (Appendix A). 

Cells expressing aNEK7 exhibited an elongated morphology, as previously reported [14]. Our data further indicate that the elongated morphology of aNEK7 cells is maintained even in the presence of TM V3 (Figure 5A), suggesting that the activation of NEK7 has a key role in driving the morphological phenotype downstream of EML4–ALK V3. Specifically, both the protrusion and total cell lengths of non-induced cells were shorter (26.22 μm and 59.80 μm, respectively) compared to cells expressing aNEK7 (43.50 μm and 83.56 μm, respectively). In the presence of either WT or TM V3, their protrusion (42.04 μm or 40.86 μm, respectively) and total cell lengths (81.94 μm or 85.42 μm, respectively) remained statistically unaltered (Figure 5B,C). In contrast, cells expressing YFP–NEK7 KD did not adopt an elongated morphology, even in the presence of the WT V3 protein (Figure 5A). The protrusion and total cell lengths of non-induced cells were the shortest (23.25 μm or 50.55 μm, respectively). When transfected with WT V3, non-induced cells exhibited longer protrusions and total cell lengths (39.05 μm and 71.95 μm, respectively) (Figure 5D,E). This was likely due to the endogenous wild-type NEK7 being recruited to microtubules by V3 and promoting the morphological phenotype. Following induction of the YFP–NEK7 KD, cells had similar protrusions and total cell lengths (24.94 μm and 53.14 μm, respectively) to non-induced cells, which were maintained even in the presence of the WT V3 protein (22.80 μm and 44.37 μm, respectively) (Figure 5D,E). Overall, our data confirm that activation of NEK7 downstream of EML4–ALK V3 drives the generation of an elongated morphology.

Eg5 is a motor protein and a direct substrate of NEK6 and NEK7. Eg5 phosphorylation by these NEK kinases is implicated in mitotic processes such as centrosome separation [26] or mitotic spindle assembly [27]. In addition, Eg5-S1033 phosphorylation by NEK7 promotes the generation of neuronal dendrites [28] as well as the morphological phenotype in cells expressing V3 [15]. This prompted us to examine the localization of Eg5 using the “knocksideways” approach. Our data indicate that Eg5 exhibits a lower co-localization with interphase microtubules (R = 0.5) in the absence, compared to the presence (R = 0.66), of EML4–ALK V3 on microtubules, suggesting that Eg5 is recruited to microtubules by FKBP–V3. Following treatment with rapamycin, Eg5 dissociated from microtubules (R = 0.57) (Figure 5F,G). Finally, Eg5 strongly co-localized with FKBP–V3 on microtubules (R = 0.67) (i.e., untreated) but not on the mitochondria (i.e., rapamycin-treated) (R = 0.22) (Figure 5H,I).

Taken together, our data confirm that Eg5 is recruited to interphase microtubules in the presence of the EML4–ALK V3–NEK9–NEK7 complex but dissociates from microtubules following the relocation of V3 to mitochondria. Importantly, Eg5 co-localizes with V3 on microtubules but not on the mitochondrial membrane. This likely prevents Eg5 phosphorylation by NEK7 and thus contributes to the reversal of phenotypic changes. 

## 4. Discussion

The EML4–ALK oncogenic fusion protein drives tumour progression in ~5% of NSCLC cases. The existence of multiple EML4–ALK variants leads to differential responses to conventional ALK inhibitors, with patients eventually relapsing and developing resistance [7,8,9]. EML4–ALK V3 is the most aggressive variant and is associated with increased metastasis in patients [10,13], which warrants the need for novel therapeutic approaches to be developed specifically against this variant. Recently, a novel pathway was described, where EML4–ALK V3 is recruited to the microtubule network together with the NEK9 and NEK7 mitotic kinases. Overall, this likely leads to the activation of NEK7 and the phosphorylation of its downstream targets such as Eg5, resulting in an increased cell length and enhanced migration [14,15]. Here, using a novel approach [16] to relocate EML4–ALK V3 from microtubules to the mitochondria, we demonstrate, for the first time, that the microtubule association of V3 is key for the generation of an elongated morphology and enhanced migratory phenotypes.

Microtubule localization of EML4–ALK is substantiated by EML4 NTD. EML4–ALK V3 is the only variant that is localized to microtubules, whereas other variants have a diffuse cytoplasmic localization due to differences in their domain organization [11,14]. Our data confirm that both the “knocksideways” FKBP–V3 and wild-type YFP–V3 used for phosphomimetic studies are localized to the microtubule network, as expected. We also show that the phosphomimetic mutants of EML4–ALK V3, mutated in sites known to promote microtubule binding, associate less with interphase microtubules compared to the wild-type V3. In fact, our data align with previous findings showing phosphomimetic EML4 mutants (S144D/S146D) being absent from interphase microtubules and phosphonull EML4 mutants (S144A/S146A) remaining on microtubules, even during mitosis [22]. Our data further highlight the significance of these key residues in the microtubule localization of EML4 and hence their use in studying the importance of the microtubule interaction of EML4–ALK. 

The “knocksideways” mechanism allows for the rapamycin-induced relocation of an FKBP-tagged protein to the mitochondria, given the presence of a mitochondrially targeted FRB domain [16,19]. By transiently transfecting Beas2B cells with the “knocksideways” FKBP–V3, we were able to induce the relocation of FKBP–V3 from the microtubule network to the mitochondria following a 5 min treatment with 200 nM of rapamycin. Interestingly, the duration of the rapamycin treatment needed to be increased in stable U2OS:FKBP–V3 cells for the relocation of FKBP–V3 to occur. In fact, the concentration and duration of treatment of rapamycin seems to vary depending on the experimental setup, as different conditions require different concentrations or durations of treatment with rapamycin to successfully induce the rerouting of the protein of interest [16,19,29,30]. Importantly, the concentration and duration of treatment of rapamycin used in this study are unlikely to have caused significant changes to the physiology of the cells, as previous findings suggest that such changes occur only at high concentrations of rapamycin (>10 ng/mL) given for long periods of time (24–48 h) [31,32]. 

Recruitment of the EML4–ALK V3–NEK9–NEK7 complex to interphase microtubules leads to the generation of an elongated morphology and enhances cell migration [14]. This novel pathway could potentially account for the increased metastasis observed in NSCLC patients with EML4–ALK V3 [10]. Using the “knocksideways” approach to relocate V3 to the mitochondria, we show that in the absence of EML4–ALK V3 from microtubules, cells do not exhibit either phenotype. Importantly, this was consistent when using the phosphomimetic mutants of EML4–ALK V3 that do not localize to microtubules. In addition, using a stable U2OS:FKBP–V3 cell line, we show that the invasive properties of V3 cells are also diminished when displacing EML4–ALK V3 from the microtubule network. We recognize that each experiment could have been performed using multiple cell lines to further strengthen our conclusions; however, this was unfortunately not feasible. Instead, several approaches were used to confirm and validate each finding.

Current findings indicate that both phenotypic changes are dependent on the presence of NEK9 and NEK7 [14]. Using cells expressing a constitutively active NEK7 or a NEK7 KD, our data confirm that the activation of NEK7 is required to drive the generation of an elongated morphology. Moreover, we show that the generation of the V3 phenotypes as a result of the NEK7 activation occurs downstream of EML4–ALK. Taken together, we can therefore hypothesise that microtubule association of V3 is required to assemble the V3–NEK9–NEK7 complex on interphase microtubules, leading to the untimely activation of NEK7 and the phenotypic observed changes.

Eg5 phosphorylation by NEK6 and NEK7 promotes mitotic progression and morphological changes in neuronal dendrites, respectively [26,28]. Recently, Eg5 phosphorylation by NEK7 was also found to be a pre-requisite for driving the elongated morphology in V3 cells. Interestingly, the enhanced migratory phenotype was found to be independent of Eg5 phosphorylation [15]. This could be suggestive of a co-operation between several NEK7 substrates for the generation of the V3 phenotypes. Hence, we can further propose that NEK7 activation, through assembly of the V3–NEK9–NEK7 complex on interphase microtubules, promotes untimely microtubule recruitment and phosphorylation of several NEK7 substrates, leading to the generation of the V3 phenotypes. Importantly, removal of V3 from microtubules might dismantle the complex, preventing the microtubule association of those substrates and their subsequent phosphorylation by NEK7 and resulting in a reversal of the phenotypes. Indeed, our co-localization studies confirmed the assembly of the V3–NEK9–NEK7 complex on interphase microtubules and recruitment of Eg5 to interphase microtubules in the presence of the complex. Furthermore, our data indicate that displacement of EML4–ALK V3 from microtubules leads to the disassembly of the complex. Future studies could examine the phosphorylation and activation levels of NEK9, NEK7 and Eg5 in response to EML4–ALK displacement from microtubules using phospho-specific antibodies against relevant residues or kinase activity assays, respectively.

Using the “knocksideways” approach, we observed that a majority of NEK7 remained in the complex with FKBP–V3 on the mitochondria in rapamycin-treated cells, whereas NEK9 and Eg5 relocated to the cytoplasm (Figure 6A). This is the first report of a NEK9-independent interaction between NEK7 and EML4–ALK V3, as NEK7 was always considered to be recruited to microtubules through interactions with NEK9 [14,24,25,26]. EML4–ALK V3 utilizes the EML4 basic region to interact and bind to the acidic E-hooks on microtubules [33]. It is likely that the following interaction allows the unstructured region between the EML4 TD and its fusion point with ALK to acquire a stable conformation, revealing a binding site for NEK9. This stable conformation may be lost upon relocation of V3 to the mitochondria, which may prevent NEK9 binding, suggesting that the binding regions of EML4–ALK with microtubules and NEK9 are close together and interdependent (Figure 6B). For this reason, a future structural characterization of the V3–NEK9 and V3–NEK7 interactions would be valuable.

ALK inhibitors are the standard treatment for ALK+ cancers with first-, second- and third-generation inhibitors already in the clinic [7]. Different variants of EML4–ALK elicit differential responses to ALK inhibitors, with V3 being the most aggressive variant and exhibiting enhanced metastasis in patients [10,34]. Regardless, most, if not all, patients eventually acquire resistance to these inhibitors and succumb to the disease [7,9]. Currently, several resistance mechanisms have been described, including mutations within the ALK catalytic domain, the activation of pathways independent of ALK catalytic activity or the upregulation of downstream components of ALK signalling pathways [35]. The EML4–ALK V3–NEK9–NEK7 pathway is an example of an ALK activity-independent pathway that may be activated in patients in response to treatment with ALK inhibitors; although, this pathway has not yet been reported in patients. Regardless, uncovering its underlying mechanisms is of utmost importance to improve patient survival. In the future, it would be interesting to examine whether the addition of peptides competing for microtubule binding with EML4–ALK V3 can lead to a phenotype reversal by displacing the protein from microtubules. If proven effective, peptide treatment may be applied in patients in response to developing resistance to ALK inhibitors. Most importantly, by targeting the interaction of EML4–ALK with microtubules instead of using inhibitors against NEK7, NEK9 or NEK7 substrates, the extent of off-target adverse events would likely be limited, as EML4–ALK is not found in non-cancerous tissues. 

Overall, our findings highlight the microtubule association of the EML4–ALK V3 oncogene as a potential future target for V3+ patients, as we demonstrate its importance in driving the elongated morphology and enhanced migratory and invasive phenotypes observed in V3 cells.

## Figures and Tables

**Figure 1 cells-13-01954-f001:**
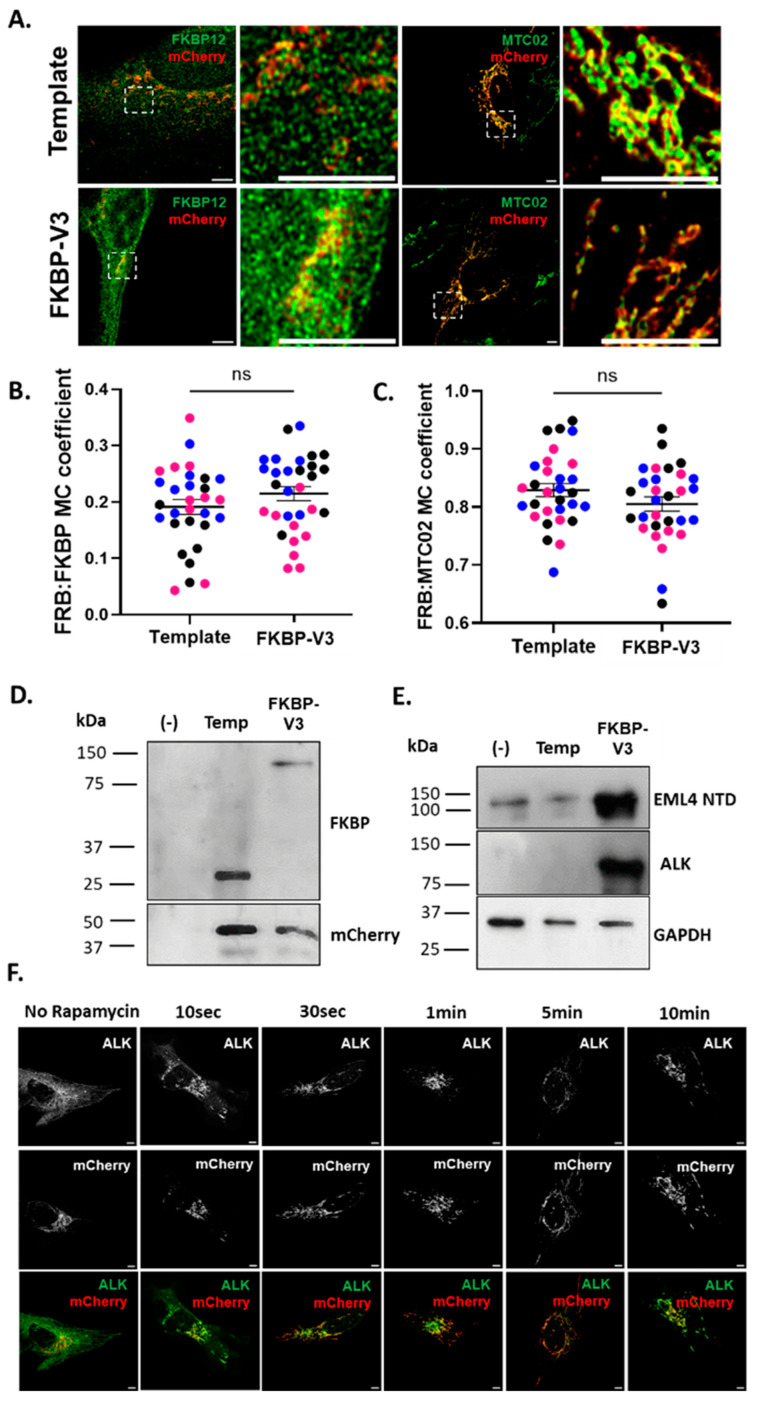
**Generation and characterisation of the “knocksideways” FKBP:EML4–ALK V3.** (**A**) Beas2B parental cells were transfected with the template bicistronic and the FKBP:EML4–ALK V3 (FKBP–V3) plasmids before being fixed and stained with antibodies against FKBP (green), mCherry (red) and MTC02 (green). Unpaired *t*-tests were performed to measure Manders’ co-localization (MC) coefficient (±SEM) between (**B**) FKBP and mCherry–FRB or (**C**) mCherry–FRB and MTC02. Each graph shows three replicates, and each replicate is colour-coded (n = 30 cells in total per sample population). Temp = template plasmid and ns = not significant. (**D**,**E**) Lysates were prepared for Western blot analyses with antibodies against ALK, mCherry, FKBP, EML4-NTD and GAPDH. Molecular weights (kDa) are indicated on the left. Blots are representative of three biological replicates (n = 3). Uncropped blots are indicated in the Appendix A. (**F**) Beas2B parental cells were transfected with the FKBP–V3 plasmid and either left untreated or treated with 200 nM of rapamycin for 10 s, 30 s, 1 min or 5 min before being fixed and stained with antibodies against ALK (green) and mCherry (red). The experiment was performed in triplicate (n = 3), and the images shown are representative of all three replicates. (**A**–**F**). Scale = 5 μm.

**Figure 2 cells-13-01954-f002:**
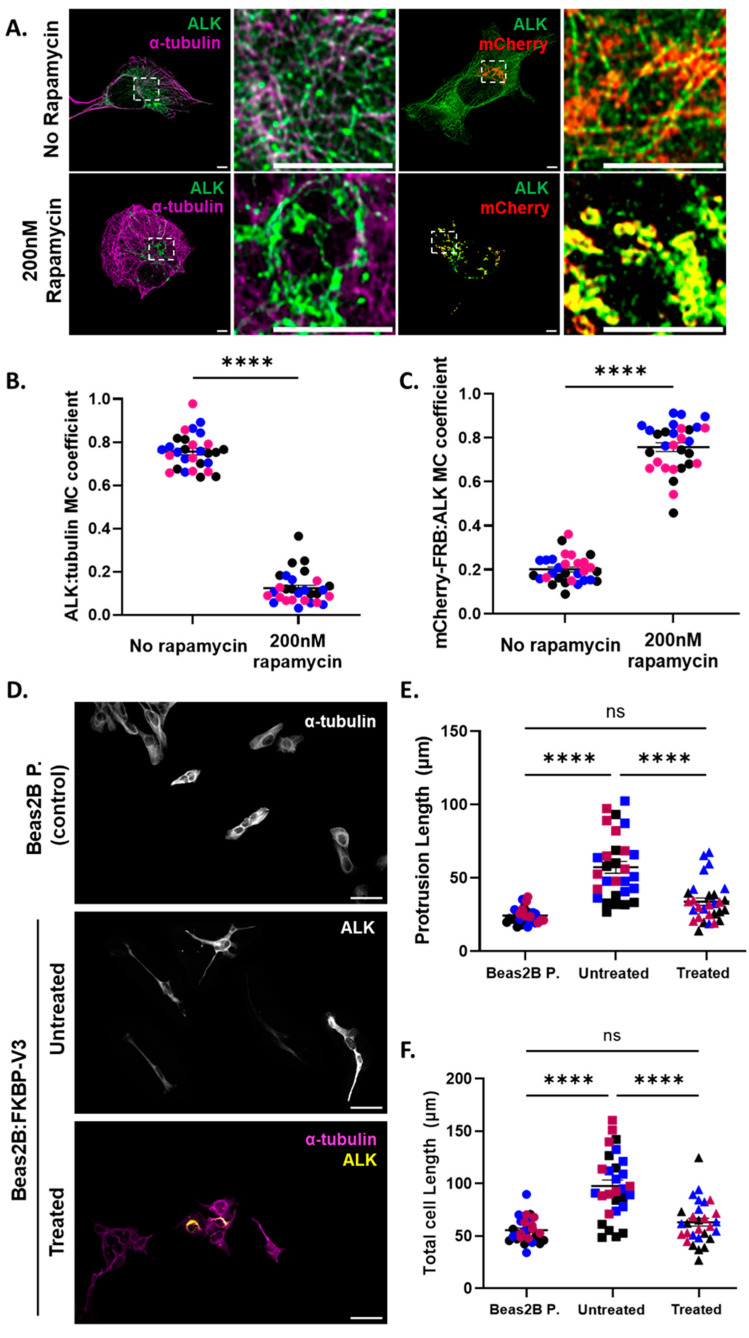
**Microtubule recruitment of EML4–ALK V3 drives the development of a mesenchymal-like cell morphology.** (**A**) Beas2B parental cells were transfected with the “knocksideways” FKBP–V3 construct and treated with 200 nM of rapamycin for 5 min before being fixed and stained for ALK (green) and α-tubulin (purple) or mCherry (red). Scale = 5 μm. Unpaired *t*-tests were performed to compare Manders’ co-localization (MC) coefficients (±SEM) between (**B**) ALK and α-tubulin or (**C**) mCherry–FRB in untreated versus treated cells. The experiment was performed in triplicate, and 10 measurements were taken from each replicate (colour-coded) (n = 30 cells in total per sample population). (**D**) Transfected cells were treated with rapamycin as above before being fixed and stained for either α-tubulin (purple), ALK (yellow) or both. Scale = 50 μm. (**E**) The protrusion (μm ± SEM) and (**F**) total cell lengths (μm ± SEM) were measured, and one-way ANOVA with Tukey’s multiple comparisons tests were performed. **** *p* < 0.0001 and ns = not significant. All experiments were performed in triplicate (n = 29 cells in total per sample population), and replicates are colour-coded. All images are representative of all replicates.

**Figure 3 cells-13-01954-f003:**
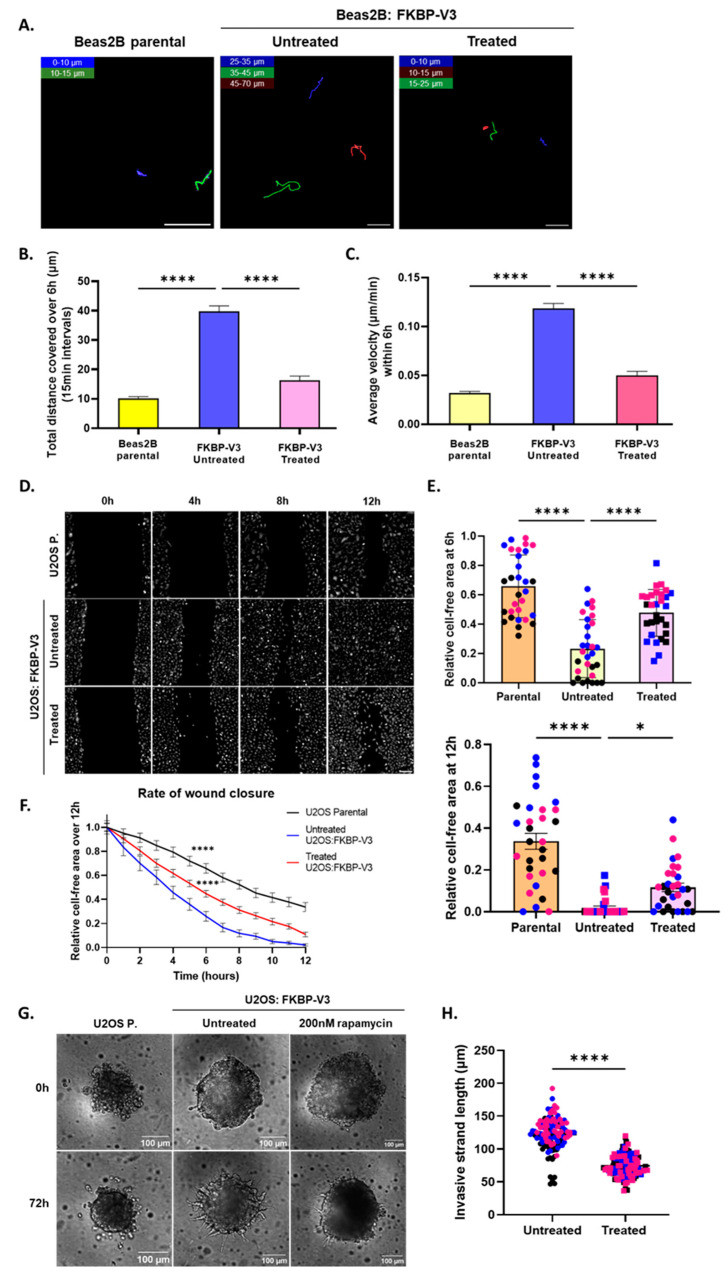
**Microtubule recruitment of EML4–ALK V3 is required for enhanced cell migration.** (**A**) Beas2B parental cells were transfected with the “knocksideways” FKBP–V3 construct and treated with 200 nM of rapamycin for 5 min before being subjected to live-cell imaging for 6 h. Scale = 50 μm. A single-cell tracking analysis was performed, and the (**B**) total distance covered (μm ± SEM) and (**C**) average velocity (μm/min ± SEM) were measured. One-way ANOVA with Tukey’s multiple comparisons tests were performed. The experiment was performed in triplicate (n = 30–40 cells in total per sample population), and measurements were taken from at least 10 cells per replicate (colour coded). (**D**) U2OS parental and U2OS:FKBP–V3 cells were allowed to reach 90–100% confluency before a wound was generated at the centre of the well. Cells were also treated with 200 nM of rapamycin for 2 h where appropriate before live-cell imaging for 12 h was undertaken. Scale = 150 μm. (**E**) One-way ANOVA with Tukey’s multiple comparisons tests were also performed to compare the cell-free area (μm^2^ ± SEM) at t = 6 and t = 12 relative to the t = 0 of U2OS parental cells. (**F**) Non-linear regression lines were drawn to determine the rate of migration, and one-way ANOVA with Tukey’s multiple comparisons tests were performed to compare the rates. The comparisons shown are against the untreated sample. The experiment was performed in triplicate (n = 30 regions in total per sample population), and each replicate is colour-coded. (**G**) U2OS parental and U2OS:FKBP–V3 cells were allowed to form spheroids in ultra-low attachment plates before embedding in Matrigel and live-cell imaging, which was performed for 72 h. Cells were also treated with 200 nM of rapamycin for 2 h where appropriate prior to embedding in 2.5 mg/mL of Matrigel. Scale = 100 μm. (**H**) At 0 and 72 h post-embedding in Matrigel, the length of 8–10 invasive strands in 5 individual spheroids per condition (n = 3 experiments) was measured (μm ± SEM) and compared between untreated and rapamycin-treated cells using unpaired *t*-tests. **** *p* < 0.0001 and * *p* = 0.0218. All images shown are representative of all replicates.

**Figure 4 cells-13-01954-f004:**
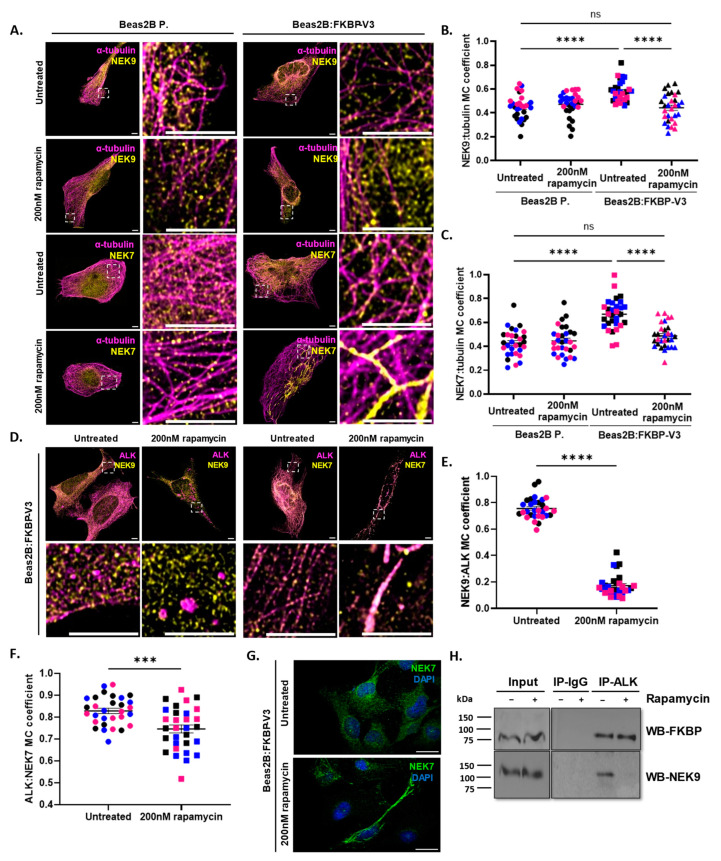
**Relocation of V3 away from microtubules results in the dissociation of the EML4–ALK V3–NEK9–NEK7 complex.** (**A**) Beas2B parental cells were transfected with the FKBP–V3 construct and treated with 200 nM of rapamycin for 5 min before being fixed and co-stained with antibodies against NEK9 (yellow) or NEK7 (yellow) and α-tubulin (purple). (**B**,**C**) One-way ANOVA with Tukey’s multiple comparisons tests were performed for comparing Manders’ co-localization (MC) coefficient (± SEM). (**D**) Beas2B cells, transfected as before, were fixed and co-stained for NEK9 (yellow) or NEK7 (yellow) and ALK (purple). (**A**,**D**) Scale = 5 μm. (**E**,**F**) Unpaired *t*-tests were performed for comparing the MC coefficients (± SEM). *** *p* = 0.0003 and **** *p* < 0.0001. Experiments were performed in triplicate (n = 30 cells in total per sample population), and each replicate is colour-coded. (**G**) Beas2B cells, transfected as before, were fixed and stained for NEK7 (green) only. Scale = 10 μm. (**H**) Lysates were also collected and prepared for immunoprecipitation using anti-ALK and anti-IgG antibodies. Immunoprecipitates were analysed by Western blotting with antibodies against FKBP and NEK9. Molecular weights (kDa) are indicated on the left. Three biological replicates (n = 3) were performed, and all images and blots are representative of all replicates. Uncropped blots are indicated in the Appendix A.

**Figure 5 cells-13-01954-f005:**
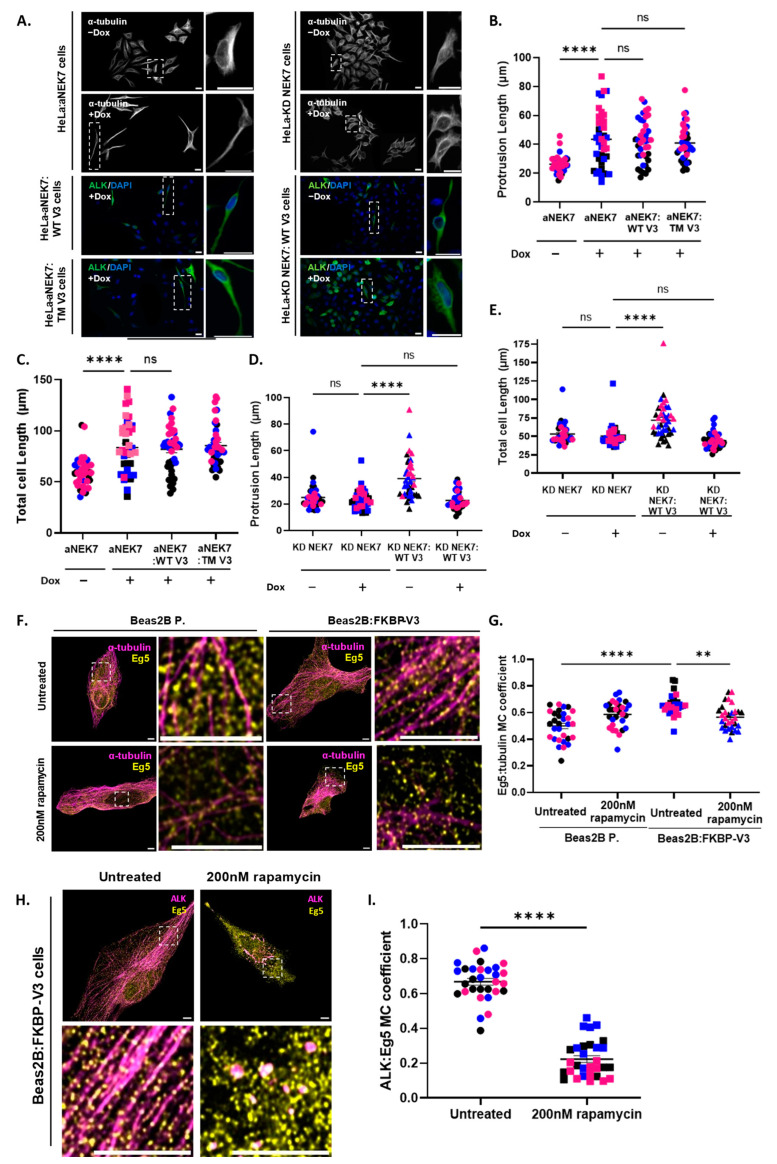
**Removal of EML4–ALK V3 from microtubules likely prevents the phosphorylation of Eg5 by NEK7.** (**A**) HeLa:aNEK7 or NEK7 KD cells were induced with 1 μg/mL of doxycycline for 72 h and transfected with the WT or TM (S134D/S144D/S146D) V3 constructs before being fixed and stained with α-tubulin or ALK (green). Scale = 25 μm. (**B**–**E**) One-way ANOVA with Tukey’s multiple comparisons tests were performed to compare protrusion lengths (μm ± SEM) and total cell lengths (μm ± SEM). (**F**) Beas2B parental cells were transfected with the FKBP–V3 construct and treated with 200 nM of rapamycin for 5 min before being fixed and co-stained with antibodies against Eg5 (yellow) and α-tubulin (purple). (**G**) One-way ANOVA with Tukey’s multiple comparisons tests were performed for comparing Manders’ co-localization (MC) coefficients (±SEM). (**H**) Beas2B parental cells were transfected with the FKBP–V3 construct and treated with rapamycin as above before being fixed and co-stained with antibodies against Eg5 (yellow) and ALK (purple). (**G**,**H**) Scale = 5 μm. (**I**) Unpaired *t*-tests were performed for comparing MC coefficients (±SEM). ** *p* = 0.0054, **** *p* < 0.0001, ns = not significant. All experiments were repeated in triplicate (n = 30–40 cells in total per sample population), and a minimum of 10 measurements were taken per replicate (colour-coded). All images are representative of all replicates.

**Figure 6 cells-13-01954-f006:**
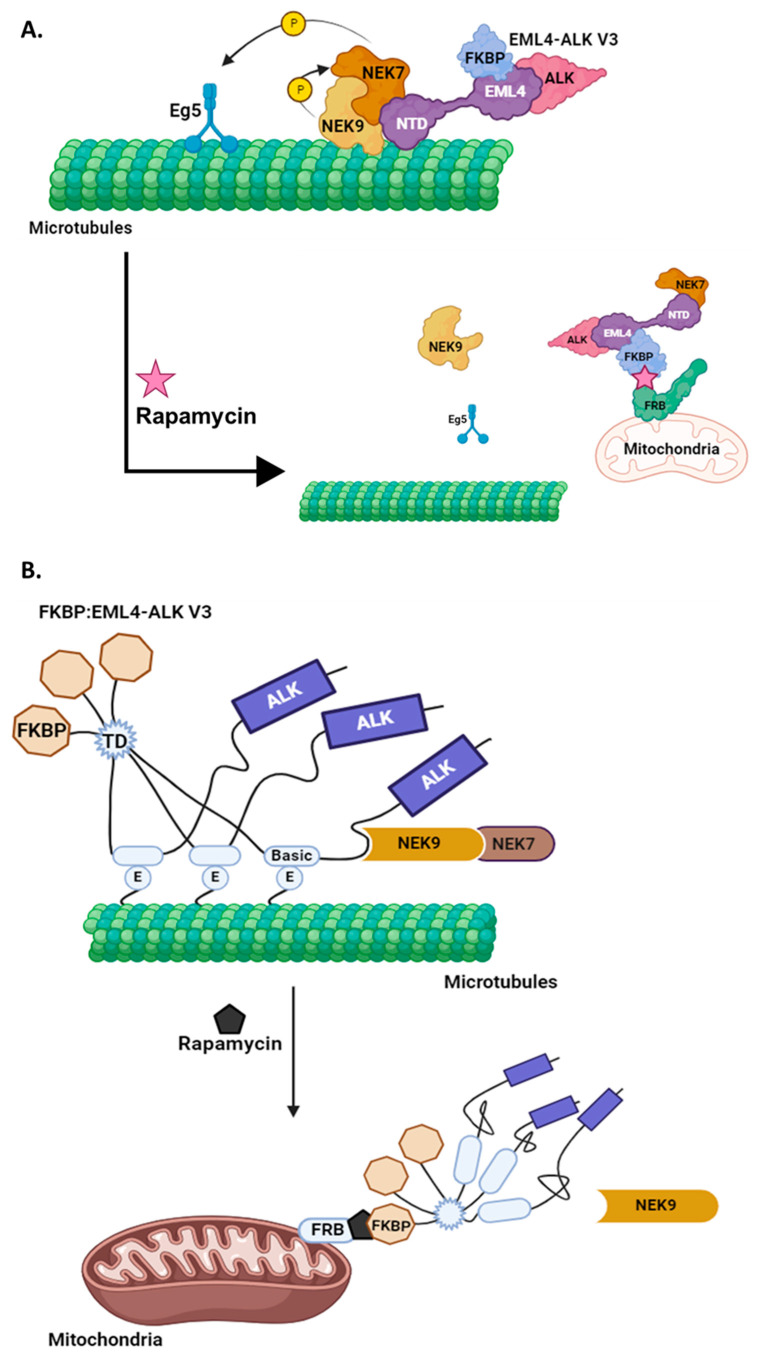
**Working models for the effects of microtubule recruitment of the EML4–ALK V3 protein on the V3–NEK9–NEK7 pathway**. (**A**) A schematic illustration demonstrating that recruitment of the V3–NEK9–NEK7 complex to microtubules leads to the phosphorylation of Eg5. Overall, this promotes a mesenchymal-like cell morphology while also enhancing the migratory potential of these cells. The addition of rapamycin leads to the relocation of V3 and NEK7 to mitochondria but not NEK9 or Eg5. This likely prevents NEK7 activation and the subsequent phosphorylation of Eg5 by NEK7, which leads to a reversal of both phenotypic changes. (**B**) A cartoon of a possible mechanism under the “knocksideways” conditions, where an interaction of EML4–ALK V3 with microtubules allows the unstructured regions between the EML4 basic region and EML4–ALK junction point to become structured, revealing a binding region for NEK9. The relocation of V3 to mitochondria following the addition of rapamycin causes that region to become unstructured again, which prevents NEK9 from binding to EML4–ALK V3.

## Data Availability

All data described are contained within this article or the Appendix A.

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
