# Peer review of "Microtubule Association of EML4–ALK V3 Is Key for the Elongated Cell Morphology and Enhanced Migration Observed in V3 Cells†"

_cells, 2024, doi:10.3390/cells13231954_

Round 1

Reviewer 1 Report

Comments and Suggestions for Authors

Papageorgiou et al. investigated the biological effects of the V3 variant of the EML4-ALK fusion, commonly associated with aggressive forms of non-small cell lung cancer (NSCLC). Their previous work demonstrated that this V3 variant localizes on interphase microtubules, interacting with the NEK9 and NEK7 kinases.

In this study, the authors employed two innovative approaches to disrupt the oncogenic fusion protein's association with microtubules. They show that the interaction of EML4-ALK V3 with microtubules is crucial for promoting both the elongated morphology and enhanced migratory behavior of cancer cells, likely contributing to increased metastatic potential. Mechanistically, they demonstrate that displacing V3 from microtubules leads to the dissociation of the EML4-ALK V3-NEK9-NEK7 complex, which in turn prevents NEK7 activation and subsequent phosphorylation of Eg5. These findings suggest a novel approach for addressing ALK inhibitor resistance, potentially offering alternative therapeutic strategies.

Overall, the manuscript is well-structured, addresses a timely and significant topic, and presents scientifically sound experiments that are likely to be of high interest to the readership.

Minor Points

Line 55-57: The authors should clarify why only the V3 variant, among EML4-ALK variants containing the tubulin-binding (TD) and basic regions necessary for microtubule localization, exhibits this specific localization pattern. This would help readers to understand the unique structural and functional aspects of V3.

Chapter 3.4: The clarity of this section could be improved. Consider simplifying the language and rephrasing sentences to enhance readability. In particular, lines 468-470 could benefit from revision for greater clarity.

Figure 4H: The quality of the immunoprecipitation-western blot (IP-WB) in this figure is suboptimal. The authors should provide clearer, more compelling evidence to support their conclusions.

Line 529-531: The statement that activated NEK7 enhances migration downstream of V3 should be either supported by experimental data or removed, as it currently lacks direct evidence.

Reviewer 2 Report

Comments and Suggestions for Authors

In this study, Papageorgiou and coworkers report  mechanistic  analyses of an aggressive variant of EML4-ALK fusion protein. Their main conclusion is that recruitment of NEK9 and NEK7 leading to activation of NEK7 by the fusion protein is responsible for inducing shape change and aggressive behavior of the cell. Overall, this is an interesting proposal and likely to be true, but the manuscript has several shortcomings.

1. In figures 2 and 3, results from treatment of wild type cells with rapamycin only is not presented (unlike figure 4). This must be corrected as rapamycin can have big impact on cellular physiology on its own.

2. The authors change the cell line a couple of times for different experiments. In fact, all of the experiments should be carried out with multiple (at least two) cell lines to assert the validity of a given observations.

3. Can we see more done with aNEK7 and kdNEK7? For example, invasion/migration assay  should be carried out. It would also be nice to see if aNEK7 can function as a 'fully-endowed' oncogene carrying out  colony formation, xenograft, etc. Such data would strongly support the conclusions.

Round 2

Reviewer 2 Report

Comments and Suggestions for Authors

It is very regretable that the authors failed or did not attempt to make adequate responses. While they try to argue around providing additional data, I would like to remind them that a story should be built on a more complete and robust data. 
